# A Cross-Species Analysis Reveals Dysthyroidism of the Ovaries as a Common Trait of Premature Ovarian Aging

**DOI:** 10.3390/ijms24033054

**Published:** 2023-02-03

**Authors:** Marco Colella, Danila Cuomo, Valeria Nittoli, Angela Amoresano, Alfonsina Porciello, Carla Reale, Luca Roberto, Filomena Russo, Nicola Antonino Russo, Mario De Felice, Massimo Mallardo, Concetta Ambrosino

**Affiliations:** 1Biogem Scarl, Institute of Genetic Research, 83031 Ariano Irpino, Italy; 2Department of Science and Technology, University of Sannio, 82100 Benevento, Italy; 3Department of Cell Biology and Genetics, College of Medicine, Texas A&M University, College Station, TX 77843, USA; 4Department of Chemical Sciences, University of Naples “Federico II”, 80138 Naples, Italy; 5Department of Molecular Medicine and Medical Biotechnologies, University of Naples “Federico II”, 80138 Naples, Italy; 6Institute Experimental Endocrinology and Oncology “Gaetano Salvatore” (IEOS-CNR), 80138 Naples, Italy; 7National Biodiversity Future Center (NBFC), 90133 Palermo, Italy

**Keywords:** cross-species analysis, ovarian dysthyroidism, premature ovarian aging (POA), chlorpyrifos (CPF), diets

## Abstract

Although the imbalance of circulating levels of Thyroid Hormones (THs) affects female fertility in vertebrates, its involvement in the promotion of Premature Ovarian Aging (POA) is debated. Therefore, altered synthesis of THs in both thyroid and ovary can be a trait of POA. We investigated the relationship between abnormal TH signaling, dysthyroidism, and POA in evolutionary distant vertebrates: from zebrafish to humans. Ovarian T3 signaling/metabolism was evaluated by measuring T3 levels, T3 responsive transcript, and protein levels along with transcripts governing T3 availability (deiodinases) and signaling (TH receptors) in distinct models of POA depending on genetic background and environmental exposures (e.g., diets, pesticides). Expression levels of well-known (*Amh*, *Gdf9*, and Inhibins) and novel (*miR143/145* and *Gas5*) biomarkers of POA were assessed. Ovarian dysthyroidism was slightly influenced by genetics since very few differences were found between C57BL/6J and FVB/NJ females. However, diets exacerbated it in a strain-dependent manner. Similar findings were observed in zebrafish and mouse models of POA induced by developmental and long-life exposure to low-dose chlorpyrifos (CPF). Lastly, the T3 decrease in follicular fluids from women affected by diminished ovarian reserve, as well as of the transcripts modulating T3 signaling/availability in the cumulus cells, confirmed ovarian dysthyroidism as a common and evolutionary conserved trait of POA.

## 1. Introduction

Ovaries are the earliest-aging organs. Their physiological aging, consisting of the gradual decline of follicle quantity and quality, is considered the driver of the same process in other organs [1]. The early decline of ovarian function occurs in around 10% of women, and it is characterized by a Diminished Ovarian Reserve (DOR) [2]. Epidemiological and experimental studies have shown that genetic and environmental factors can accelerate the reduction of ovarian lifespan and healthspan [3,4]. The genetic background likely influences ovarian susceptibility to environmental stressors, such as endocrine disruptors (EDCs), which can modulate the endocrine pathways involved in preserving ovarian health, inclusive of the ones active along hypothalamus–pituitary–gonadal (HPG) axis [5,6,7].

Thyroid hormones (THs), Triiodothyronine, and Thyroxin (T3 and T4, respectively, from now on) regulate ovarian healthspan and lifespan, although their favorable role is still debated. They can promote ROS production and chronic inflammation, processes that reduce longevity and ovarian lifespan [8,9,10,11,12].

T4 and T3 levels are tightly regulated by feedback mechanisms along the hypothalamic–pituitary–thyroid (HPT) axis. The hypothalamus secretes Thyrotropin-Releasing Hormone (TRH) that induces the release of Thyroid-Stimulating Hormone (TSH) from the pituitary. The latter stimulates the production of T4 and T3 in the thyroid gland, which, in turn, inhibits TRH and TSH synthesis. A small fraction of T3 is produced by the thyroid gland in vertebrates; the majority derives from the conversion of T4 into T3, mediated by the iodothyronine deiodinases (DIOs) within the cells of peripheral organs. Hence, T3 levels are customized peripherally by the life stage-specific expression of the DIOs, distinguished in activating enzymes (DIO1 and DIO2, transforming T4 in its more active form T3, active when the T3 levels are low in the organ) and inactivating enzymes (DIO3, transforming both into their biologically inactive metabolites, active when T3 level are too high within the organ) [13,14]. TH transporters modulate the local availability of THs. Lastly, TH tissue-specific signaling and transcriptional activity depend on an array of receptors (TRs) and associated proteins present in the cell. The customization of T3 signaling at the peripheral level is evolutionarily conserved along with the mechanisms regulating ovarian activity [15].

The role of the local TH metabolism and signaling in ovaries has been explored in mammals, especially in rodents and humans. TH transporters, receptors, and the deiodinases (dio1, dio2, dio3a, and dio3b) have been reported to modulate Zebrafish development. Also in humans, the mRNA and protein levels of these markers have been reported in more cellular components of the follicles and at different maturation stages [4].

Recently, we have shown that environmental factors such as diet and pesticides promote POA, revealing novel cellular mechanisms of POA [16]. Chlorpyrifos (CPF) is an organophosphorus pesticide retrieved in environmental matrices and human samples [17,18]. It has been found in cordon blood, implying that the exposure can start at the most vulnerable age [19]. Currently, epidemiological studies have been focused on its neurotoxicity and carcinogenesis [20,21]. Its ability to affect reproduction and promote follicle atresia has been described in mice and zebrafish [22,23,24]. In both species, CPF exposure has been associated with a decrease in circulating estrogen levels and derangement of other hormones active along the HPG axis. Interestingly, CPF’s role in promoting dysthyroidism has been suggested in several reports in both rodents and zebrafish [25,26].

Our study aims to characterize ovarian T3 dysthyroidism as an evolutionarily conserved target of genetic and environmental factors affecting ovarian health and aging. Although such analyses have been complicated by the role of circulating T3 (cT3) in modulating the expression of genes regulating its organ availability, signaling, and transcriptional activity, we provide evidence that ovarian T3 (oT3) metabolism and signaling pathway play a role in the preservation of ovarian lifespan and healthspan. The latter has been evaluated by measuring a platform of well-characterized and recently suggested molecular biomarkers of ovarian aging, summarized in Table 1. Indeed, the reported data evidence the disarrangement of T3 metabolism and signaling in vertebrates suffering from ovarian dysfunctions, due to genetic or environmental factors, from teleost to humans.

## 2. Results

### 2.1. Ovarian Dysthyroidism in Genetically Different Mice Exposed to Different Diets

In our previous work, we investigated the role of both genetic background and diets on POA onset in mice [16]. In order to characterize the relation between circulating levels of fT3 (cfT3, from now on) levels and genetic assets, we took advantage of samples already collected in that study [16]. We measured circulating levels of fT3, by ELISA assay, in 8-month-old females of two different strains: C57BL/6J and FVB/NJ (Figure 1a,b). The analyses evidenced a slight increase in cfT3 in C57BL/6J females vs. age-matched FVB/NJ mice (Figure 1a,b) fed with standard diet. Feeding with American and Ketogenic diets promoted its reduction in both strains, whereas the Mediterranean diet had the same effect only in FVB/NJ females (Figure 1a).

We have already reported a slight difference between the two strains in ovarian aging and that such differences were exacerbated by the adopted diet [16]. Briefly, by RT-qPCRs analyses of well-established and novel biomarkers of ovarian aging, we reported that C57BL/6J females had “younger ovaries” than the age-matched FVB/NJ females and that the Mediterranean diet had a beneficial effect on ovarian health only in C57BL/6J [16]. Although we had no samples to measure the intra-ovarian level of fT3 (ofT3, from now on) by ELISA, we monitored the ovarian T3 signaling by investigating the level of T3-regulated transcripts and of the T3 metabolizing enzymes and receptors. The data evidenced that the transcript of the activating enzyme *Dio1* was more abundant in FVB/NJ vs. C57BL/6J females (Appendix A). No other significant difference was detected for *Dio2* and *Dio3* mRNAs (Appendix A), as well as for the transcripts of the TH receptors (THRs, from now on) and a T3-responsive gene (*Spot14*). Then, we investigated the interplay between genetic and environmental factors in modulating cfT3 and ovarian T3 signaling, as above. As shown in Figure 1a,b, American and Ketogenic diets caused a decrease in cfT3 in both strains, whereas the Mediterranean diet exerted this effect only in FVB/NJ. Then, we analyzed the expression levels of the deiodinases. *Dio3* mRNA levels (Figure 1c,e) were increased in both strains fed the Mediterranean diet. Moreover, a trend towards upregulation was detected in C57BL/6J females fed any diet vs. mice fed standard mouse chow (Figure 1c), which was consistent with the trend towards the decrease in *Dio2* mRNA in the same samples (Figure 1d). Interestingly, we found a decrease in *Dio2* mRNA in both strains fed with the diets that we previously reported to have a strain-dependent negative impact on ovarian health and aging: Mediterranean diet for FVB/NJ and Ketogenic diet for C57BL/6J (Figure 1f,d). Overall, the data suggested that C57BL/6J ovaries presented a hyperthyroid status independent from cfT3, which was also present in FVB/NJ fed Mediterranean diet. These results are corroborated by the T3 receptor alpha and beta expression analyses, which have been shown to be regulated by T3 administration [27]. In fact, both transcripts were reduced in C57BL/6J females fed a Ketogenic diet and showed a trend towards downregulation in females fed the American diet (Figure 1g,h), whereas they were both induced in FVB/NJ females fed with these diets (Figure 1i,j). Finally, to understand the real state of ovarian T3 signaling, we assessed the transcriptional levels of *Spot14*, a canonical T3/THR responsive gene [28,29,30]. In agreement with the expression profile of THRs under different diets, we found its transcript decreased in C57BL/6J fed Ketogenic diet.

Taken together, the data suggest that genetics is the main factor in determining the difference between strains in response to stressors. Despite the fact that the American and Ketogenic diets promoted cfT3 reduction in both strains, only FVB/NJ was able to customize ovarian T3 signaling by increasing the expression of THRs.

### 2.2. Ovarian Dysthyroidism Is a Trait of POA Induced by Developmental and Lifelong Exposure to CPF in Zebrafish

Disease mechanisms conserved in evolutionary distant vertebrates are often relevant to humans [31]. Zebrafish is considered a model for several human diseases, including endocrine and reproductive ones [32]. Since we already conducted a study aimed to characterize the effect of environmentally relevant exposure to chlorpyrifos (CPF) on the metabolism and signaling of THs in zebrafish, we took advantage of collected samples to characterize ovarian dysthyroidism in females. Their reproductive dysfunctions [24,26] were revealed by investigating the impact of CPF on the reproductive health of the exposed females breeding them with unexposed males as described in the Material and Methods section (M&M, from now on). Briefly, AB wild-type zebrafish embryos were exposed starting from 6 h post fertilization (6 hpf) until adulthood (180 days post-fertilization, dpf) to CPF 30 nM and CPF 300 nM, adding it to fish water (Appendix A). Then, both unexposed (Vehicle, from now on) and exposed females were mated with unexposed males. Fertilized embryos were obtained from all the mating pairs, and we observed a dose-dependent reduced ability to produce fertilized eggs in exposed females (Figure 2a). Furthermore, an increase in abnormal segmentation was also evidenced (Appendix A). Next, we assessed molecular markers of ovarian health and aging of granulosa cells (*amh*, *inhibins*, etc.) and oocytes (*gdf9*, *bmp15, oct4*, etc.), whose expected regulation is summarized in Table 1. Accordingly, *amh* (Figure 2b) and *inhbb* (Appendix A) mRNA levels were decreased in the ovaries of females exposed to 300 nM CPF. Moreover, *inha* transcript was reduced only when exposed to 30 nM CPF (Appendix A). No major changes were observed in *inhba* and *fst* expression (Appendix A). CPF exposure did not impact oocytes’ health since the ovarian transcript levels of *gdf9, bmp15, oct4, zp2,* and *sycp1* remained unchanged (Appendix A). Then, we verified the expression of ncRNAs whose impaired expression has been previously associated with ovarian aging [33]. We measured exclusively the expression of *dre-mir-143*, *dre-mir-145,* and *gas5* since the others do not have orthologs in zebrafish. As expected, up to what is summarized in Table 1, both *dre-mir-143* and *dre-mir-145* were increased at both doses (Figure 2c,d), whereas *gas5* was reduced only at the highest dose (Figure 2e). The data pointed to a shortened ovary lifespan in the exposed females, which was consistent with the increased Foxo3a phosphorylation (a well-characterized marker of ovarian aging in mammals, Figure 2f,g), verified by Western blotting [34,35]. Concordantly, we also detected reduced telomeres length at all doses, assessed by qPCR (Figure 2h) [36,37].

Then, ovarian dysthyroidism was characterized by measuring ovarian fT3 (o-fT3, from now on) by ELISA assay in ovarian homogenates prepared from the same samples. As shown in Figure 3a, it was reduced in females exposed to CPF 30 nM, whereas a trend towards the decrease was evidenced at higher doses. Concordantly, we detected a decrease in T3 responsive transcripts as *igfbp1a* mRNA (Figure 3b) and *esr1* mRNA (Figure 3c), observed only at the high dose for the former and at all doses for the latter [38,39]. However, esr1 protein levels did not change significantly (Appendix A). In addition, no major effects on *dio1* mRNA were detected (Figure 3d), whereas *dio2* mRNA levels were reduced only at the higher dose (Figure 3e), leading to ovarian hypothyroidism. In agreement with the hypothyroid status of the ovaries exposed to high-dose CPF, the transcripts of the deactivating deiodinases (*dio3a, dio3b*) were reduced following the same dose pattern (Figure 3f,g). 

Remarkably, the data indicated that ovaries from zebrafish exposed to CPF were hypothyroid and presented both molecular and phenotypic signs of POA.

### 2.3. Ovarian Dysthyroidism Is a Trait of POA Induced by Developmental and Lifelong Exposure to CPF in Mice

To characterize ovarian dysthyroidism as a conserved trait of POA in rodents, we took advantage of samples and data collected in a previous study aimed to investigate the effect of CPF exposure from conception to adulthood on the thyroid gland [40]. Briefly, mice were exposed from E0.5 to PND21 via dams fed medicated food with low dose CPF (1 mg/kg/day) and high dose CPF (10 mg/kg/day) or left untreated (CTRL). The offspring was directly exposed from weaning (Appendix A) to sacrifice (6 months) [40]. Reproductive ability was assessed in 10 females from each exposure group. As shown in Table 2, 1 mg/kg/day CPF females did not produce pups, whereas 15 pups were delivered by 10 mg/kg/day CPF females vs. the 39 delivered by the CTRL group. The mating was stopped right after the first delivery in order to count the placental button at sacrifice. The results, reported in Table 2, suggested a difficulty to conceive since the earlier stages because only 7 placental buttons were retrieved in the 1 mg/kg/day CPF group. Such effect was poorly pronounced in 10 mg/kg/day CPF, which showed an increase in miscarriage when compared to the CTRL group. Hormonal imbalance along the HPG axis was assessed, and results were detailed in Appendix A. Relevant to our results were the increased levels of serum 17-β estradiol (E2, from now on) in 1 mg/kg/day CPF vs. CTRL females and the altered expression of *Fsh-β* and *Lh-β* in the pituitary, which was reduced and increased in a dose-dependent manner, respectively (Appendix A). As summarized in Appendix A, we further observed a decrease in cfT3 in 10 mg/kg/day CPF females vs. the CTRL group corroborated by the increase in *Tsh-β* in the pituitary.

The occurrence of POA was investigated in the ovaries of 6-month-old females exposed to CPF by assessing the levels of the somatic and gametal molecular markers, also investigated in zebrafish. *Amh* mRNA was reduced at all doses of exposure (Figure 4a), whereas *Inha* and *Inhbb* were increased (Figure 4b,c). No major change was detected in *Inhba* (Appendix A). Furthermore, reduced expression of *Gdf9* (Figure 4d) and *Bmp15* (Figure 4e) at both doses was detected, suggesting an oocyte impairment, confirmed by the decreased expression of *Oct4*, *Sycp1* (Appendix A), and *Zp2* in 10 mg/kg/day CPF ovaries (Figure 4f). Concordantly, a trend toward upregulation was evidenced for both *miR143* and *miR145*. However, the former was increased in a statistically significant manner solely in females exposed to 1 mg/kg/day CPF (Figure 4g,h). Furthermore, *miR505* (Figure 4i) was increased in the ovaries of females exposed to 1 mg/kg/day CPF, whereas *Gas5* was reduced at all conditions (Figure 4j). Finally, we verified the length of telomeres, which was shortened in the exposed ovaries regardless of the dose vs. CTRL (Table 2), is also a marker of ovarian senescence [36,37]. We also examined ovarian apoptosis, which was confirmed by the increase in *Bax/Bcl-2* transcripts ratio, although only in 1 mg/kg/day CPF ovaries vs. CTRL (Appendix A).

Then, we evaluated CPF exposure effects on ovarian customization of T3 signaling. We detected a decrease in several T3-regulated transcripts, such as *Spot14* (Figure 5a) and *Cpt1a* (Figure 5b). Since we did not have the possibility to measure the ovarian T3 level in this experimental setting, we investigated the expression of other T3-regulated genes, such as *Sdha* and *Ndfus3*, which were both found to be reduced (Appendix A) [41]. Since T3 negatively regulates *Cyp19a1* transcription in rodent granulosa cells [42], we detected the expected increase in *Cyp19a1,* although only in 10 mg/kg/day CPF ovaries (Figure 5c). Then, because we monitored *esr1* transcript levels in zebrafish, which were found to be reduced (Figure 3c), we evaluated Esr1 protein by IHC in the ovaries from CTRL and CPF-exposed mice. The latter was decreased in a dose-dependent manner (Figure 5d–f”). Such data were further corroborated by the analysis of granulosa cell (GCs) proliferation, which is positively regulated by T3 [43]. Concordantly, we observed a dose-dependent reduction of GCs proliferation by staining the ovaries of CTRL- and CPF-exposed females with an antibody against Ki-67 (Appendix A).

These data suggested ovarian hypothyroidism in the exposed ovaries, which was further confirmed by the *Dio1* and *Dio2* expression profiles (Figure 5g,h). Both mRNAs were increased in 10 mg/kg/day CPF ovaries vs. CTRL. No major changes were retrieved in the expression of *Dio3* (Appendix A). *Thrb* mRNA was reduced in females exposed to 1 mg/kg/day CPF and increased at higher doses (Figure 5i), whereas *Thra* mRNA showed a decreasing trend at any dose (Appendix A). In addition, an increasing trend was detected for the *Oatp1c1* transporter (Appendix A). No major change was observed in the *Mct8* transcript (Appendix A).

### 2.4. Ovarian Dysthyroidism Is an Evolutionary Conserved Trait of POA Retrieved Also in Humans

Overall, the data confirmed that hypothyroidism is an evolutionary conserved and common trait of POA due to genetic and/or environmental cues. Therefore, we hypothesized that it could also be involved in human POA. This suggestion was verified by taking advantage of samples collected and used in a previous study aimed to validate *MIR143/145* and other molecular markers as novel biomarkers of POA in women suffering the diminished ovarian reserve (DOR), a major trait of POA [16]. We verified that our samples were from euthyroid women. Indeed, two of them, one among infertile women not suffering from DOR and one among DOR patients, were subjected to therapy for hypothyroidism. We determined the levels of fT4, fT3, and E2, by ELISA, in follicular fluids (FF) from CTRL and DOR women (Figure 6a–c). As shown, the fT4 was not different, whereas fT3 and E2 were both reduced in FFs from DOR samples. The RNA prepared from Cumulus Cells (CCs) of the same patients was used to determine the expression of DOR molecular markers, also assessed in the animal models. As expected, *AMH* mRNA (Figure 6d) was reduced, whereas *MIR505* (Figure 6e) was induced, according to already published data [16,44]. We measured the expression levels of deiodinases in CCs, retrieving the reduction of *DIO2* mRNA in DOR samples (Figure 6f). Lastly, we confirmed hypothyroidism by monitoring the levels of T3-regulated transcripts in CCs. In agreement with the T3-reduction in FFs from DOR patients, we found an increased level of *CYP19A1* mRNA (Figure 6g). Consistent with the published data, we found a correlation between the *DIO2* transcript level and the number of MII oocytes (Figure 6h).

## 3. Discussion

Aging is a physiological process that has been associated with endocrine dysfunction involving both the thyroid and gonads [45]. Several reports have discussed the association of aging with changes in thyroid function; however, the reciprocal association is less explored because of the complexity of THs synthesis occurring in the thyroid gland and in the periphery [4,46]. Thyroid hormones are major players in the aging of different organs, including ovaries. The latter point is strongly debated, likely due to the way TH levels and function are assessed, which relies on the measurements of the circulating levels of T4 and T3 [47,48,49]. This type of analysis provides only a partial view since T3 signaling is customized locally by a tissue/cell-specific array of enzymes, transporters, and receptors. Therefore, dysthyroidism can be locally established or compensated even independently on the circulating THs levels.

With this study, we aimed to validate ovarian dysthyroidism as a common and evolutionary conserved trait of POA, assessing it in distant phylogenetic vertebrates, such as zebrafish and rodents, in which both genetic and/or environmental factors are known to contribute to POA onset. Finally, we confirmed our hypothesis by mirrored analyses in human samples.

We have already reported that genetic background and diets co-contribute the rise of POA in mice [16]. Here, we further investigated the possibility of dysthyroidism in these samples. Our investigation suggests that genetic background and diet slightly influence POA and ovarian dysthyroidism; however, their interaction promotes both. In fact, a decreased oT3 signaling was evidenced in ovaries from C57BL/6J fed Ketogenic diet—a condition promoting POA-and was increased in ovaries from FVB/NJ—a condition protecting from POA. Interestingly, it has been reported that the Ketogenic diet induces hypothyroidism in humans as well as in mice because such a diet mimics a fasting condition that is known to promote hypothyroidism [49]. Although we have not directly assessed it, the balance of deiodinases and THRs transcript levels suggest that it is compensated differently at the peripheral level depending on the strain (Figure 1). The decrease in *Thra* and *Thrb* mRNA levels, resulting in the inhibition of T3 signaling, has been suggested as a common trait of POA induced by different factors in mice [50]. The participation of TH-receptors in the regulation of ovarian function has also been observed in mouse models carrying a whole-body homozygous deletion of the TRs genes [51,52]. Our data are in agreement with previous studies showing the age-related decrease in *THRB* expression in other cell types (PBMC) as a result of the increased methylation of its promoter [53]. Further studies are necessary to characterize the molecular mechanism leading to their inhibition in ovaries. Consistently with previously published reports, we also evidenced differences depending on genetic background for *Dio1* mRNA levels, whereas the diets have less of an impact [54]. The data are in contrast with the reported role of environmental factors modulating the expression of deiodinases in the liver [55]. We were surprised by such results because the decrease in THs is considered a favorable factor in physiological aging and because the increase in TH availability and signaling seems to be protective in damaged ovaries [56]. This discrepancy might depend on the modulation of the estrogen signaling by THs. Indeed, T3 cooperating with FSH, regulates the expression of the aromatase (Cyp19a1) and of ER alpha, playing an important role in folliculogenesis in maintaining both the ovarian somatic cell phenotype and ovarian plasticity [57]. Therefore, activation of the ovarian E2 signaling would be fundamental to developing a protective response and might be dependent on ovarian T3 availability and signaling. Further experiments are required to address the role of genetic factors.

To further validate this evidence and investigate its suitability in different vertebrates, potentially in humans, we took advantage of experiments conducted to characterize the effects of developmental and lifelong exposure to CPF on the HPT axis. In our models, mice and zebrafish, exposure to CPF impaired fertility and promoted aging. In agreement with previous reports, we found that 1 mg/kg/day CPF, by promoting the increase in circulating T3, had a deeper impact on female fertility than the higher dose (10 mg/kg/day CPF) [58,59]. These phenotypic data are consistent with the molecular analyses of the markers of ovarian aging. The similarity in the expression pattern for *Dio1, Dio2, Spot14,* and *Cpt1a* between 1 mg/kg/day CPF and 10 mg/kg/day CPF exposed females points to ovarian hypothyroidism in both conditions (Figure 5). However, these results are in contrast with the expression pattern of *Cyp19a1* at 1 mg/kg/day CPF. As a matter of fact, hypothyroidism has been reported to induce *Cyp19a1* mRNA levels in rat ovaries following the PTU treatment [60]. Interestingly, the authors further reported a decrease in the number of pups and the implantation sites, which we report in our study as well [60]. The conflicting effects on *Cyp19a1* expression detected in 1 mg/kg/day CPF and 10 mg/kg/day CPF might depend on the complex modulation of *Cyp19a1* expression via FSH signaling that also involves OCT4 modulation by T3 [61]. *Oct4* was similarly regulated, as expected [62], at all doses, whereas *Fsh-β* mRNA was reduced in the pituitary from 10mg/kg/day CPF females but not in 1 mg/kg/day CPF females due to the increase in circulating E2 in the latter [63]. Since it has been reported that T3 lowered *Cyp19a1* activity required activated FSH signaling, which is reduced in our condition, and such activity cannot be exerted in our model, and another signaling can promote it [64]. 

Ovarian hypothyroidism is corroborated by the reduction of the proliferation of granulosa cells (Ki-67 staining) and of ER alpha evidenced by IHC [65]. Noteworthy, the aging in the exposed ovaries was confirmed by telomere shortening [66]. All the above-reported parameters agreed with the deranged expression of well-established (*Amh*, *Gdf9*, *Bmp15,* and other ovarian peptides) along with novel described biomarkers (*miR143, miR145*, *miR505,* and *Gas5*). Newly, here we propose the role of *miR505* as a marker of ovarian aging in both mice and humans. It can target the mitogen-activated protein kinase kinase kinase 3 (MAP3K3) regulating the AKT/NF-kB pathway [67,68]. Both pathways are involved in ovarian aging [69,70].

Relevant to our aim were the consistent results observed in zebrafish, an evolutionary distant vertebrate. Although zebrafish has asynchronous ovaries, their development, gamete maturation, and the signaling pathways involved in the regulation of both processes are conserved across species [71]. Furthermore, T3 signaling regulation/function and the mechanisms governing its homeostasis are preserved [72]. We confirmed that exposure to CPF impairs fertility and promotes hypothyroidism, given the reduced levels of ovarian T3. Differently from mice, the detected hypothyroidism has been suggested by the inhibition of *Dio3* transcripts beyond the increment of *Dio1* and *Dio2* mRNAs. On the other hand, the inhibition of *Thra* has been conserved in both animal models at the higher dose of CPF, whereas *Thrb* was regulated in an opposite manner. The decreased expression of vitellogenin (data from another paper under consideration) only in these samples suggests an impact on the synthesis of estrogen that would need other analyses considering that T3 cooperates with FSH in modulating *Cyp19a1* transcription. Consistently with the results obtained in mice, Esr1 (IHC) staining was reduced in the hypothyroid ovaries. This implies a dysregulation of the estrogen signaling promoting the telomere shortening for the exposed mouse females.

Regarding analyses in humans, our data agreed with previous studies showing that T3 levels were positively correlated to the number of MII oocytes [73,74] (Figure 6h). Newly, we describe that such a condition is strictly determined by the activity of deiodinase enzymes in the cumulus cells.

Taken together, the data confirmed that inadequate oT3 availability and signaling is a common trait of POA that remains poorly investigated. We suggest that ovarian hypothyroidism can promote ovarian aging because of its negative impact on estrogen signaling.

## 4. Materials and Methods

### 4.1. Mouse Treatment to Different Diets and CPF

All the procedures regarding mouse exposure to different diets (C57BL/6J and FVB/NJ) and CPF (CD1) have already been reported [16,40]. For the former, C57BL/6J and FVB/NJ female obtained from The Jackson Laboratory (Bar Harbor, ME) were exposed to different diets as described in previous studies (ID number 0182-2013, 0339-2016) [16,75]. To induce POA, CD1 dams (outbred strain, 8 mice/treatment group) were exposed 7 days before mating at 1 (mg/kg/day) or 10 (mg/kg/day) to CPF by medicated food. The offspring was exposed to the dams from gestational day 0 (GD 0) until the weaning, and then, they were directly exposed. Animals were sacrificed at 6 months, for blood and organ collection, by carbon dioxide inhalation [40]. All animal experiments were performed in accordance with the European Council Directive 2010/63/EU following the rules of the D.Lvo 116/92 (ID number 25-10), and the procedures were approved by the Ethical committee named CESA (Committee for the Ethics of the Experimentations on Animals) of the Biogem Institute. The number of mice enrolled in the study was established by executing a G-Power analysis, which was required in preparing the documents to obtain the authorization from the Italian Ministry of Health, that fix the parameters to use (α = 0.01; 1 − β = 0.85; δ = 4.12).

### 4.2. Zebrafish Husbandry and Treatment

Adult fishes (AB line) were maintained according to standard procedures on a 14-h light/10-h dark lighting cycle at 28 °C. Animal experiments were performed in accordance with the European Council Directive 2010/63/EU, and procedures were approved by the Italian Minister of Health (IMH, ID number 78-17). The number of mice enrolled in the study was established by executing a G-Power analysis, which was required in preparing the documents to obtain the authorization from the Italian Ministry of Health, that fix the parameters to use (α = 0.01; 1 − β = 0.85; δ = 4.12) As previously described [24], zebrafish eggs at 6 h post-fertilization (hpf), were randomly collected and placed in separate glass Petri dishes in 100 mL of fish water containing CPF (30 and 300 nM). Then, they were maintained in an incubator at 28 °C. The exposure solutions were renewed daily, and fishes were examined under a dissecting microscope for morphological evaluation. Those arrested in development or malformed were discarded. At 10–15 days post-fertilization (dpf), larvae were transferred in Stand Alone (Tescniplast), adapted for toxicology treatment, and the exposure continued until adulthood (180 dpf).

### 4.3. Human Samples

After obtaining informed consent, Follicular Fluids (FF) samples were collected from twelve patients previously described in Cuomo et al., (2018) [16].

### 4.4. Hormones Detection

Circulating fT3 (cfT3) and estradiol (cE2) were measured in serum samples (*n* = 5, per treatment group) by Gas Chromatography–Mass Spectrometry (GC-MS). The liquid–liquid extraction procedure was used to prepare the samples for GC-MS analysis of the hexane supernatant, as already reported [76]. Ovarian T3 levels (oT3) were determined by ELISA assay, following the manufacturer’s instructions [24]. Briefly, free T3 (fT3) levels were measured in homogenates of adult zebrafish ovaries (*n* = 5, per treatment group) prepared in RIPA buffer: 50 mM Tris (pH 7.4), 150 mM NaCl, 0.1% SDS, 0.5% Na-deoxycholate, Nonidet P-40, protease and phosphatase inhibitor mixture (Sigma-Aldrich, St. Louis, MO, USA), using the tissue lyser instrument. After centrifugation (10 min, 5000× *g* at 4 °C), the supernatants were collected and stored at −80 °C until hormone measurement by the ELISA kit (Diametra kit: Estradiol, DKO003, sensitivity 8.7 pg/mL; Free T3, DKO037, sensitivity 0.05 pg/mL; Free T4, DKO038, sensitivity 0.05 ng/dL) or used for Western blotting analyses. In the follicular fluids (FF), according to the manufacturer’s instructions was measured the 17-β-Estradiol (fE2) (Diametra kit: Estradiol, DKO003, sensitivity 8.7 pg/mL).

### 4.5. RT-qPCR Analysis

Total RNA from mouse (ovaries, pituitary) and zebrafish (ovaries) (*n =* 5, per treatment group) was isolated with TRIzol reagent (Invitrogen, Waltham, MA, USA) according to the manufacturer’s instructions. Reverse transcription (RT) and qPCR were accomplished using the QuantiTect Reverse Transcription Kit (Qiagen, Hilden, Germany) and Fast SYBR Green Master Mix (Applied Biosystems with Applied Biosystem QuantStudio 7 Flex System, (Waltham, MA, USA)), respectively. Primer sequences are listed in the Appendix A (see Appendix A). RT-qPCR analyses were preceded by the determination of the levels of three different internal controls *(β-actin*, *Gapdh,* and *Tubulin* for ovaries mouse; *Gapdh* for pituitary mouse; and *β-actin*, *elf1a,* and *tubaI* for zebrafish), under different exposure conditions to verify their stability. Data were normalized by the level of internal control *β-actin* for ovaries from mice and *tubaI* for zebrafish expression as a result of the stability test [77,78]. To avoid genomic DNA contamination, we designed primer sequences spanning an exon–exon junction. Experiments were performed in triplicates. The 2^−∆∆Ct^ method was used to calculate relative expression changes.

### 4.6. miRNA Isolation, cDNA Synthesis, and RT-qPCR for miRNA Detection

Total miRNAs were isolated from mouse and zebrafish ovaries using TRIzol reagent (Invitrogen) according to the manufacturer’s instructions. For mice, according to the published protocol, Reverse transcription (RT) was performed using the QuantiTect Reverse Transcription Kit (Qiagen) [16]. The RT reaction was conducted using specific stem-loop primers designed for each miRNA, as previously reported [79]. Real-time PCR was conducted as described above. The conditions were set as previously reported by Yang et al., (2014). RT-qPCR analysis was performed in triplicate. Data obtained were normalized to the relative expression of reference gene U6 small nuclear RNA. The 2^−ΔΔCt^ method was used to calculate relative expression changes with a control group as a reference point. The same protocol was used for the detection of zebrafish miRNAs. Primer sequences are listed in the Appendix A (see Appendix A).

### 4.7. Telomere Length Detection

Ovarian DNA was isolated by cutting the organs into small pieces and incubating them overnight (O/N) at 60 °C in Lysis buffer ((Tris pH 8.0 (10 mM), NaCL (100 mM), EDTA pH8.0 (10 mM), SDS (0.5%)) (750 µL), and proteinase K (10 mg/mL). Then, 250 µL of NaCl (6M) was added, and samples were centrifuged (10 min, 10,000× *g* at 4 °C). The supernatants were collected in new tube. In total, 500 µL of isopropanol was added, and the samples were centrifuged (20 min, 10,000× *g* at 4 °C). After several washes with cold Ethanol 70%, the pellet was resuspended in sterile water. Average telomere length was measured from total genomic mouse DNA by using a qPCR method previously described [80,81]. The premise of this assay is to measure an average telomere length ratio by quantifying telomeric DNA with specially designed primer sequences and dividing that amount by the quantity of a single-copy gene. For internal control, the acidic ribosomal phosphoprotein PO (*36B4*) gene has been used for gene-dosage studies in mice. For zebrafish, telomere length detection was conducted up to published protocols, and *c-fos* was used as internal control [81,82]. The PCR conditions were set at 95 °C (for 10 min) followed by 35 cycles of data collection at 95 °C (for 15 s), with 52 °C annealing (for 20 s), followed by extension at 72 °C (for 30 s). Primer sequences are listed in the Appendix A (see Appendix A). The average of these ratios was reported as the average telomere length ratio (ATLR).

### 4.8. Western Blotting Analysis

To prepare proteins, frozen ovaries (*n* = 5/group) were lysed in RIPA buffer (50 mM Tris (pH 7.4), 150 mM NaCl, 0.1% SDS, 0.5% Na-deoxycholate, Nonidet P-40, protease and phosphatase inhibitor mixture (Sigma) with the tissue lyser according to a published protocol [26]. The following antibodies were used to detect zebrafish proteins: Foxo3a (Cell signalling Tech., 2497, 1:1000), p-Foxo3a (Cell signalling Tech., 9466, 1:1000), Estrogen Receptor alpha (esr1, Santa Cruz, sc-8002, 1:1000); B-actin (Cell Signalling, 1:3000) was used to normalize data. The secondary antibodies used were anti-rabbit (G21234, 1:2000) and anti-mouse (G21D40, 1:2000) (Life Technologies, Carlsbad, CA, USA).

### 4.9. Hematoxylin and Eosin Stining and Immuno-Histochemistry (IHC)

For microscopy, mouse ovaries were fixed in formalin and embedded in paraffin. Sections were stained with Hematoxylin and Eosin (Sigma-Aldrich, St. Louis, MO, USA) according to the manufacturer’s instructions. Immunohistichemistry was performed on 5 μm ovary sections from 3 samples/group. Briefly, sections were deparaffinized and rehydrated ((Xylene 2 × 3 min), (100% Ethanol 2 × 3 min), (95% Ethanol 2 × 3 min), blocking of endogenous peroxidase (1 mL 35% H_2_O_2_ in 100 mL of Methanol, 70% Ethanol 2 × 3 min, Water, Sydney, Australia). The antigen retrieval was conducted by boiling for 20 min in Sodium Citrate Buffer (10 mM Sodium Citrate, 0.05% Tween 20, pH 6.0) and the antibody against Esr1 (1:200) Santa Cruz, sc-8002) and Ki-67 (1:500) Novus biologicals, NB110-89717) were incubated O/N at 4 °C in PBS-BSA (1%). A secondary antibody was used (1:1000) for 1 h at room temperature. After several washes, detection with DiAminoBenzidine (DAB, SK-4100, Vector Lab., Newark, NJ, USA) was used according to the manufacturer’s instructions. Sections were mounted on coverslips with Eukitt (BioOptica, Milano, Italy). Samples were imaged on a Zeiss Axioplan 2 microscope.

### 4.10. Statistical Analysis

Statistical analyses were performed using the Prism 5.0 software (GraphPad Software, La Jolla, CA, USA). Student’s *t*-test was used. Probability *p*-values below 0.05 were considered significant and indicated respectively with the symbols (*). Unless otherwise indicated, we considered a minimum of five animals for experiments. Fold change (FC) values were calculated as the ratio between average results in treated and control samples. The results are expressed as the mean ± standard deviation of independent animals. The strength of linear association between pairs of variables was determined by the Spearman correlation coefficient. Spearman’s rank correlation, which looks for any monotone relationship between two variables, was used to determine the relation of well-known ovarian aging parameters, TH metabolism, signaling genes (*DIO2* and *CPT1A*), and miRNAs expression. The calculations were performed in R using the cor.test function, and the corrplot package was used to plot the correlogram.

## 5. Conclusions

The imbalance of T3 availability/signaling in the ovary might represent an underestimated mechanism of POA induced by genetic and environmental stressors. Our results suggest that ovaries are an exception to the paradigm that a mild reduction of THs is a biomarker of healthy aging. Such paradigm needs a re-evaluation since it does not consider that T3 availability and signaling are customized locally altering pathways playing a major role in preserving ovarian health and “youth” as estrogen signaling and their crosstalk. The design of innovative therapeutic strategies should account for this point and contemplate the use of other animal models besides rodents. Our results suggest the big potential of comparative analyses in evolutionary distant vertebrate models as a successful approach to dissecting mechanisms of ovarian aging.

## Figures and Tables

**Figure 1 ijms-24-03054-f001:**
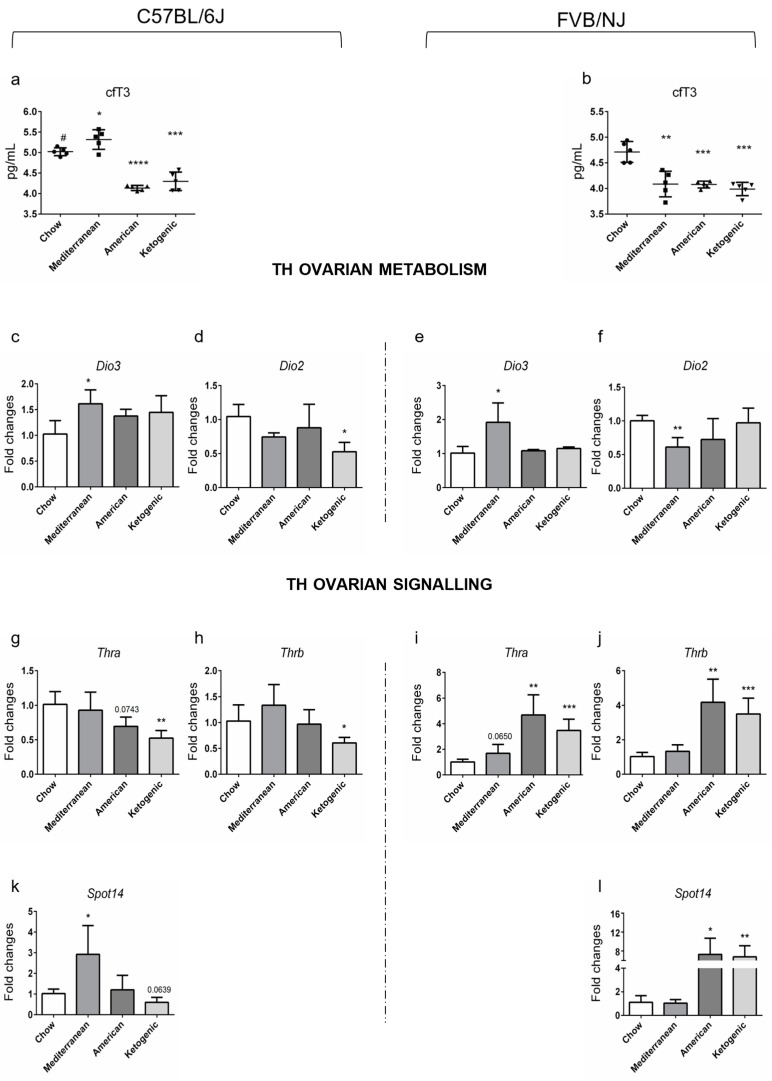
T3 levels changes in C57BL/6J and FVB/NJ mice fed different diets. (**a**,**b**) The dietary effects on circulating T3 (cfT3) levels were determined by ELISA assay (*n* = 5/group). Significant differences are indicated with * *p* < 0.005; ** *p* < 0.01; *** *p* < 0.001, **** *p* < 0.0001 using Student’s *t-*test (**c**–**l**). The mRNAs implicated in THs metabolism (*Dio2* and *Dio3*) and signaling (*Thra*, *Thrb,* and *Spot14*) were detected by RT-qPCR. Data are reported as the ratio between mRNA content in different diets and control groups normalized to β-actin. Data are mean ± s.d. with five animals per group. Significant differences are indicated with * *p* < 0.05; ** *p* < 0.01, and *** *p* < 0.001 using Student’s *t*-test. Student’s *t-*test for different relative to the cfT3 between C57BL6/J and FVB/NJ strain is indicated with # *p* < 0.05.

**Figure 2 ijms-24-03054-f002:**
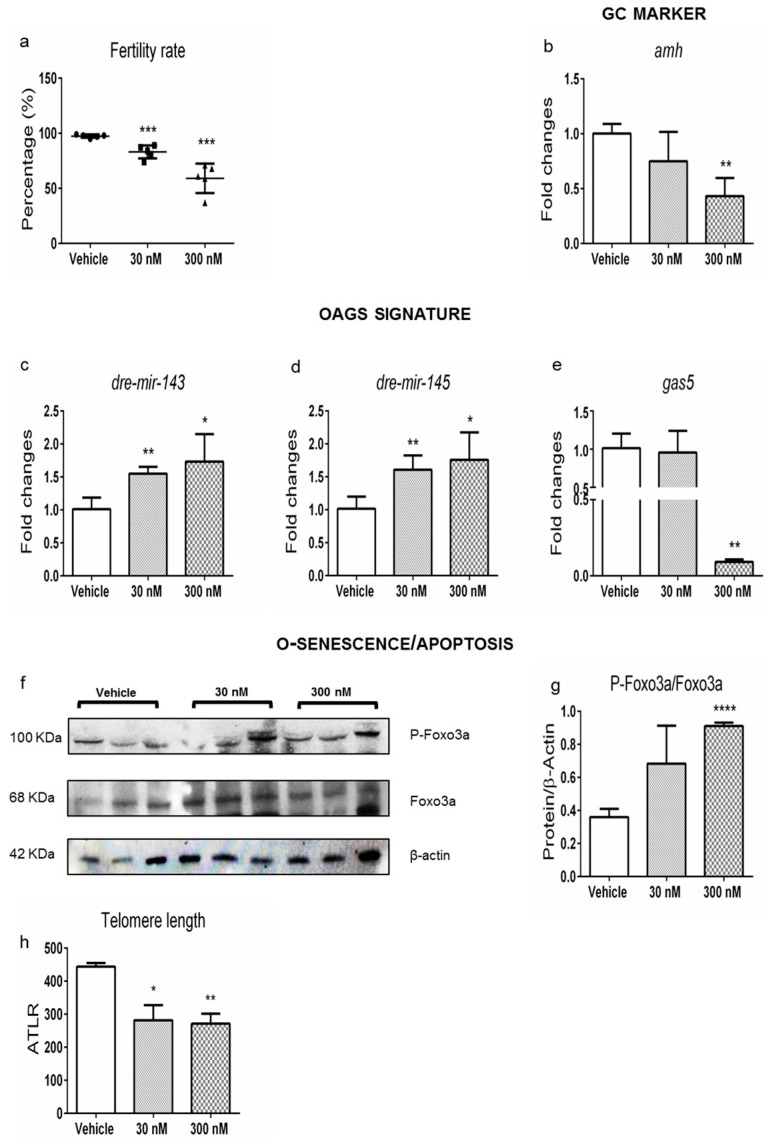
Developmental and lifelong exposure to CPF in zebrafish ovaries promotes POA. (**a**) Graph represents the fertilization percentage estimated by counting the number of fertilized eggs obtained from five independent matings involving zebrafish females exposed to CPF vs. females not exposed (Vehicle). (**b**) Granulosa cell markers (*amh*) were detected by RT-qPCR. (**c**–**e**) OAGS genes (*dre-mir-143*, *dre-mir-145,* and *gas5*) were verified by RT-qPCR. (**f**,**g**) Representative Western blot analysis showing the level of Foxo3a/P-Foxo3a protein following CPF treatment (*n* = 3/group). (**h**) Telomere length was measured from total genomic ovaries DNA by using a qPCR. Data were obtained normalizing using *tubaI* for mRNA, β-actin for proteins, and *U6* for miRNA). Data are mean ± s.d. with five animals per group. Significant differences are indicated with * *p* < 0.05; ** *p* < 0.01, *** *p* < 0.001, and **** *p* < 0.0001 using Student’s *t*-test.

**Figure 3 ijms-24-03054-f003:**
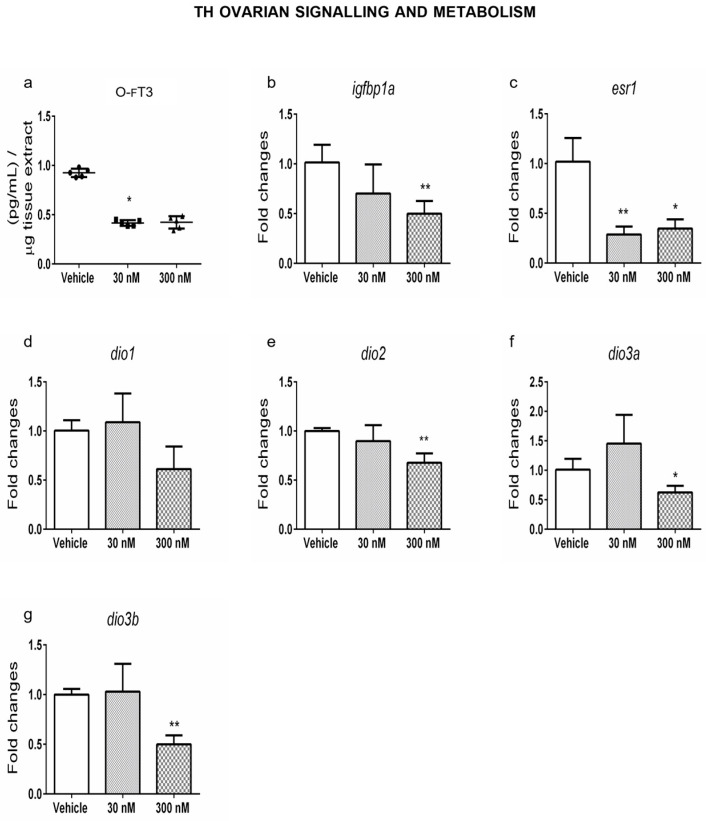
Exposure to CPF modulates THs levels and metabolism in zebrafish ovaries. (**a**) Ovarian fT3 levels (O-fT3) were measured by ELISA assay in adult ovaries from Vehicle and exposed females (*n* = 5 ovaries/group), as described in M&M section. (**b**–**g**) Levels of the mRNAs of T3 responsive genes (*igfbp1a* and *esr1)* and enzymes involved in THs metabolism (*dio1*, *dio2, dio3a,* and *dio3b)* in ovaries of zebrafish exposed to CPF. RT-qPCR tests were performed on five biological samples (*n* = 5 ovaries/group). Data are reported as fold change values calculated as a ratio between average relative gene expression in exposed and control ovaries after normalization on *tubaI* mRNA. Significant differences are indicated with * *p* < 0.05 and ** *p* < 0.01 using Student’s *t*-test.

**Figure 4 ijms-24-03054-f004:**
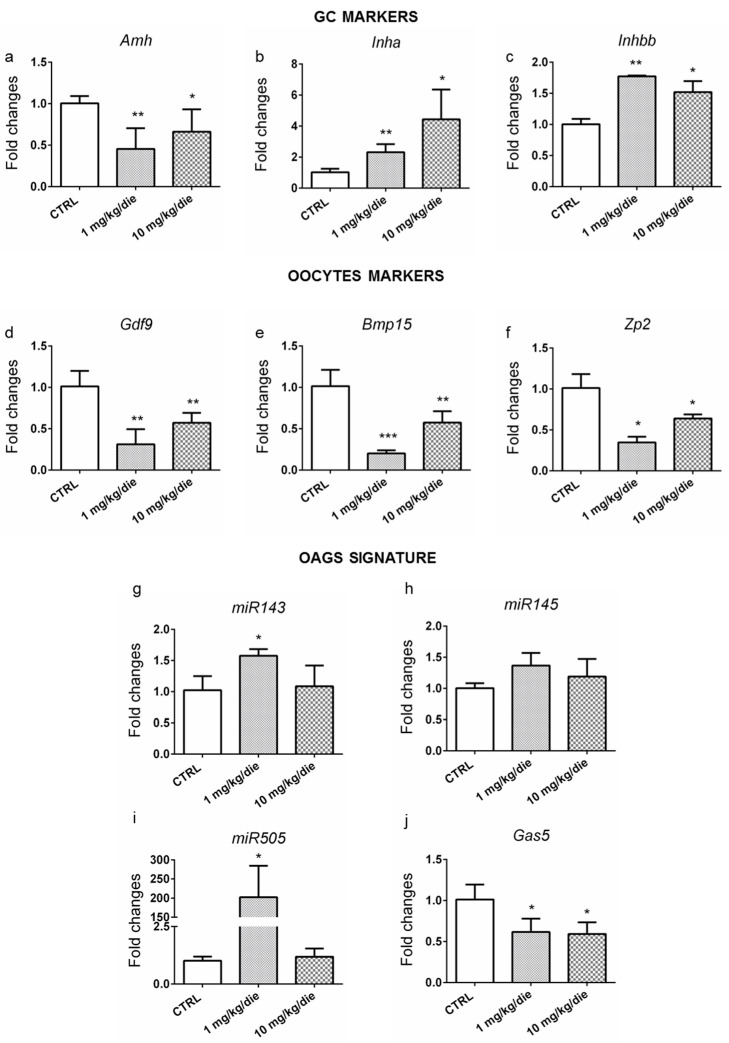
Developmental and life-long exposure to CPF affects OAGS in mice. (**a**–**f**) RT-qPCR analyses of the levels of markers of granulosa cells (*Amh, Inha,* and *Inhbb*) and markers of oocytes (*Gdf9, Bmp15,* and *Zp2*). (**g**–**j**) OAGS genes (*miR143*, *miR145, miR505,* and *Gas5*) were verified by RT-qPCR. RT-qPCR tests were performed on five biological samples (*n* = 5 ovaries/group). Data are reported as fold change values calculated as a ratio between average relative gene expression in exposed and control ovaries after normalization on *β-actin* mRNA (*U6* for miRNA). Significant differences are indicated with * *p* < 0.05, ** *p* < 0.01 and *** *p* < 0.001 using Student’s *t*-test.

**Figure 5 ijms-24-03054-f005:**
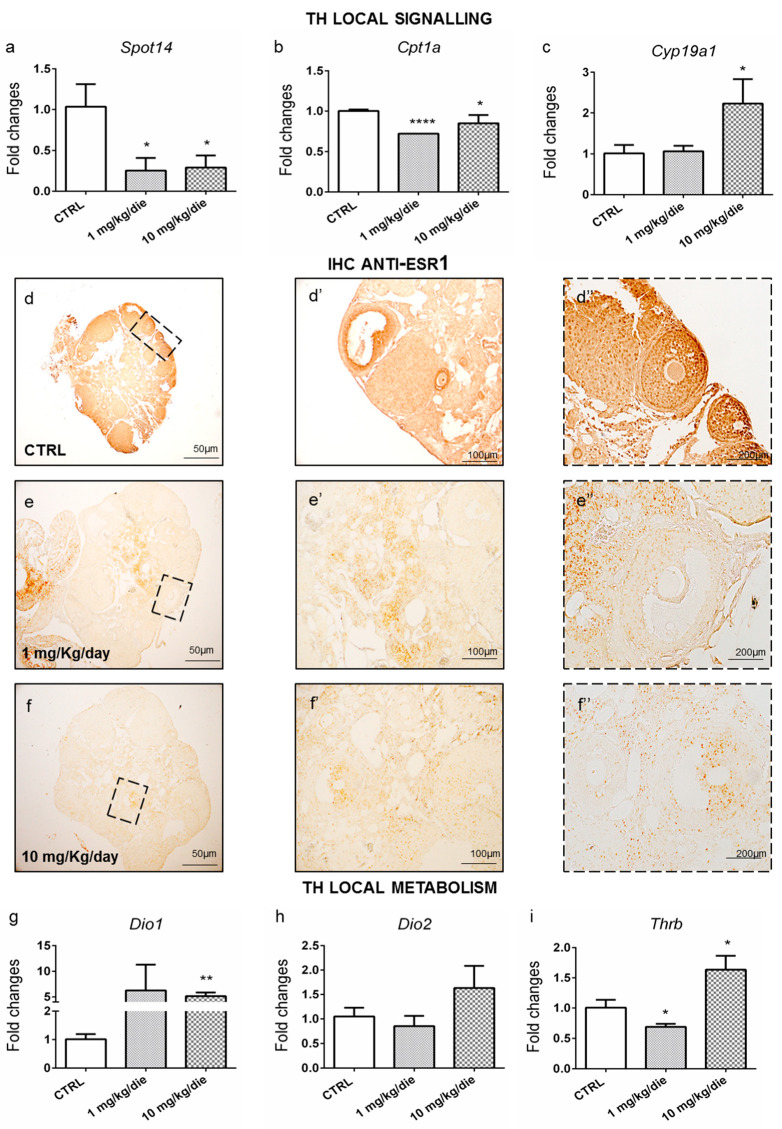
Mice developmentally and long-life exposed to CPF exhibit ovarian TH signaling and metabolism alterations. (**a**–**c**) RT-qPCR analysis of the levels of T3 responsive transcripts (*Spot14, Cpt1a,* and *Cyp19a1).* (**d**–**f”**) Staining for Esr1 on mice ovaries sections (5×, 10×, 20× magnification), showing the alteration of Esr1 level in exposed groups (*n* = 3 ovaries/group). (**d**–**d”**) Staining in CTRL groups. (**e**–**e”**) Staining in exposed group to 1 mg/kg/day. (**f**–**f”**) Staining in exposed group to 10 mg/Kg/day (**g**–**i**) TH inactivation and activation enzymes (*Dio1* and *Dio2)* and *Thrb* receptor expression were analyzed by RT-qPCR (*n* = 5 ovaries/group). Data are reported as fold change values calculated as a ratio between average relative gene expression in exposed and control ovaries after normalization on β-actin mRNA. Significant differences are indicated with * *p* < 0.05, ** *p* < 0.01 and **** *p* < 0.0001 using Student’s *t*-test.

**Figure 6 ijms-24-03054-f006:**
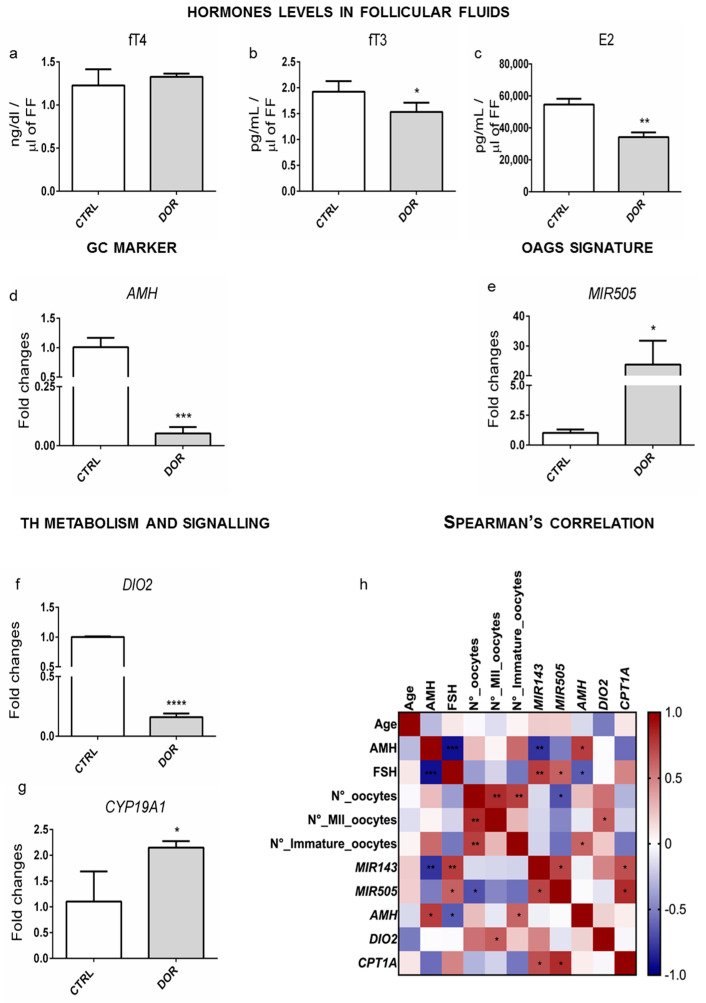
TH variation in FF and CCs of women with DOR. (**a**–**c**) Hormones levels (fT4, fT3, E2) detected by ELISA assay in FF from DOR-affected and healthy women. (**d**,**e**) *AMH* mRNA and *MIR505* levels were tested by RT-qPCR. Data are reported as the ratio between mRNA/miRNA content DOR and control groups normalized to *GAPDH/U6*. (**f**,**g**) Expression of the genes (*DIO2* and *CYP19A1*) detected by RT-qPCR. (**h**) Spearman’s rank correlation is indicated in color depth (number ranger from −1 to +1). Data are reported as fold change values calculated as a ratio between average relative gene expression in 5 pz per control group and 7 pz DOR-affected. Significant differences are indicated with * *p* < 0.05, ** *p* < 0.01, *** *p* < 0.001, and **** *p* < 0.0001 using Student’s *t*-test.

**Table 1 ijms-24-03054-t001:** Schematic representation of the marker of tissue thyroid hormone status and ovarian health in hypothyroidism and premature ovarian aging.

	Expected Marker Regulation	
Function	In Hypothyroidism [4]	In Ovarian Aging [16]
TH ovarian metabolism	cT3 ↓	cT3 ↓
	*Dio1*/*Dio2* ↑	*Dio1/Dio2* (?)
	*Dio3* ↓	*Dio3* (?)
TH ovarian signalling	*Thra, Thrb* ↓	*Thra, Thrb* (?)
	*Spot14* ↓	*Spot14* (?)
Granulosa cell markers	*Amh* (?)	*Amh * ↓
	*Inha* (?)	*Inha * ↓
	*Inhbb* (?)	*Inhbb* (-)
Oocytes cell markers	*Gdf9 * ↓	*Gdf9 * ↓
	*Bmp15 * ↓	*Bmp15 * ↓
	*Oct4* (?)	*Oct4 * ↓
	*Sycp1* (?)	*Sycp1 * ↓
OAGS markers	*MiR143* (?)	*MiR143 * ↑
	*MiR145* (?)	*MiR145 * ↑
	*MiR505* (?)	*MiR505* ↑
	*Gas5* (?)	*Gas5 * ↓

Symbol: ↑, upregulation; ↓, downregulation; (-) no change; (?) no information. Noteworthy, the regulation of the major part of the molecular marker of ovarian aging in hypothyroid status has not been reported before and was shown in this paper for the first time.

**Table 2 ijms-24-03054-t002:** Ovary healthspan and lifespan alterations in mice exposed to CPF.

	Ovarian Healthspan and Lifespan				
Mouse Exposed to CPF	Number of Dams (n°)	Percentage (%)	Pups (n°)	Placental Buttons (n°)	Telomere Length (ATLR)
CTRL	5/5	100%	39	42	4225.8 ± 1005.9
1 mg/Kg/day	5/10	30%	0	7	2638.2 ± 155.3 *
10 mg/Kg/day	5/10	40%	15	34	2410.9 ± 613.2 *

The table reports the number of dams enrolled, the percentage of successful breeding, the number of newborns (pups), the number of placental buttons, and the telomere length detected by qPCR. Data are mean ± s.d. with five animals for group. Significant differences are indicated with * *p *< 0.05 using Student’s *t*-test.

## Data Availability

Not applicable.

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
