# Peer review of "A Cross-Species Analysis Reveals Dysthyroidism of the Ovaries as a Common Trait of Premature Ovarian Aging"

_ijms, 2023, doi:10.3390/ijms24033054_

Round 1

Reviewer 1 Report

In the manuscript "A cross-species analysis reveals ovarian dysthyroidism as a common trait of premature ovarian aging", Colella et al present a study in which they investigated the Thyroid Hormone T3 level, features of ovarian health, and signatures of Premature Ovarian Aging (POA) using zebrafish, mouse and human data. They reported ovarian dysthyroidism as a trait of POA and environmental factors (diet in mouse and lifelong exposure to CPF in zebrafish) can contribute to it. Overall, the rationale for their study is well defined. However, I have some concerns as outlined below:

1. Authors used many different markers like Dio1/2/3, Thra/b, Spot14, Amh, Gdf9, Inhibins, miR143/145, Gas5 and more, it is very difficult for the readers to understand and interpret the results involving them. Can the authors make a table or schematic of the relationship of these genes and what is the expectation of the changes in the situation of hypothyroid status?

2. In figure 1, authors mentioned “a slight increase of cfT3 in C57BL/6J females vs age-matched FVB/NJ mice (Figure 1a, b)”, is this comparing chow control in C57BL/6J and FVB/NJ? If so, what is the p-value? The p-value for this comparison is not indicated in the figure.

3. In figure 1a-b, American and Ketogenic diets promoted its reduction of cfT3 in both strains, but C57BL/6J and FVB/NJ show changes in opposite directions in figure 1g-I, what is the interpretation? In this case, the thyroid level and the ovarian health is not corelated?

4. If “ncRNAs whose impaired expression has been previously associated with ovarian aging”, why the ncRNAs dre-mir-143, dre-mir-145 and gas5 show changes in different directions in figure 2c-e. and figure 4g-j. For example, dre-mir-143, dre-mir-145 are increased while gas5 is decreased. Is this result expected?

5. In the results, some of the changes reported by the authors are not statistically significant. Authors should be more careful about the language.

Minor points:

1. The panel numbers of figure S2 are not correctly referred in the main text line 155-160. For example, in line 159, figure S2f-j should be figure S2e-i.

2. In line 154, “inhbb (Figure S2b) mRNA levels were decreased in ovaries from females exposed to 300 nM CPF.” This result is not in figure S2.

3. Authors refer to Table S3 in line 217, should be Table S2 in the supplementary files.

4. In line 260, authors refer to Figure 5c for esr1 transcript level, which is not shown in the figure.

5. Typo in line 314 beginning and line 548 beginning.

6. The figure legends and axis labels in figure 1-3 are a little blur and not easy to read.

Reviewer 2 Report

In the present study, the authors assessed the T3 signalling/metabolism and markers of ovarian function in several species. They also test the effects of different diet and CPF. The authors aimed to characterize the ovarian T3 dysthyroidism that affects ovarian health and aging. However, some critical questions need to be addressed.

Major

1. Scientific concepts. Some words in the manuscript are misleading and puzzling.

1) Does “ovarian dysthyroidism” in the title mean that the dysthyroidism is attributed from ovaries?

2) Does ovarian T3 mean that the T3 is secreted form ovaries? The study has validated that T3 signaling (Thr, Dio, Spot14 etc.) existed in ovaries, so what is the source of T3 (ovaries or thyroids)?

3) What is the definition of “premature ovarian aging” as I never see this statement before. Does it equal “ premature ovarian insufficiency” or premature ovarian failure”? In addition, it is also different from DOR (line 291).

2. Study design.

1) From lines 75-76 and 318-320, the authors aimed to reveal how T3 dysthyroidism affected ovaries. So they should model dysthyroidism animals first, and then assess the ovarian functions. But in this study, all models are based on POA animals, which is on contrary to their aims.

2) The Chlorpyrifos (CPF) exposure is used to model zebrafish and mice. However, it is suggested that CPF can lead to both POA and dysthyroidism (line 71-74). So the study is more like testing the effects of CPF on ovaries and thyroids, rather than investigate the relationship of these two organs.

3. The resolution of images needs to be improved.

4. Detailed methods.

1) Since the models and treatments are critical elements in this study, some relevant comprehensive information (group, does, duration, etc.) should be fully provided. Only mentioning reference 16 is not adequate.

2) Line 462, the ELISA kits (DKO003 and DKO037) are provided for human samples, why can be used for zebrafish?  

3) In Supplementary Table 1, why do you choose three internal reference (β-actin, tubaI, Gapdh) for different species?

Round 2

Reviewer 2 Report

Thank you for the authors response. However, many answers still need more evidence to support.

1. It seems that the difference between POA and POI is the FSH level, but how to distinguish the phenoma in animal experiments? And please provide robust references about the definition POA.

2. The study did not provide direct information for ovarian aging, as AMH and E2 are not sufficient. Please supplement the results of follicle counting and FSH about mice and zebrafish. Chlorpyrifos is not that common to induce ovarian aging, so associated data is critial.

3. Is there paper reporting the T3 production by ovaries? Although ovarian cell express Dios, it does not mean that they secret T3, because many data are based on RNA levels. In fact, the hormones be from the circulation and stay in ovaries.

4. Line 455, the mice type is wrong.

Round 3

Reviewer 2 Report

1.As I said earlier, the definition of POA is vague and not often used, so more robust citation will increase the credibility. However, the authors did not provide direct evidence for POA, but rather explained the concept of Early Ovarian Ageing, which might also indicate that the “POA” is not well recognized.

2.The authors considered that “POA is also called Early Ovarian Ageing”. Nevertheless, in reference 1, Kate et al. described that “No standardised diagnostic criteria of EOA have been established, nor consensus over the nomenclature”, so it is inappropriate to explain POA by using another vague concept. In addition, Kate et al. said “Early Ovarian Ageing may be a more appropriate term than premature ovarian ageing, which implies a more advanced or severe process leading to POI”, and it suggests that POA is different from EOA.

In reference 2-4, I cannot find description that explain POA, or compare POA with POI.

3.Since the concepts of POA and POI are used in clinical practice, how to demonstrate that these animals are POA models rather than POI models without FSH levels?

4.Please added the synthesis function of ovaries in the introduction section.

Round 4

Reviewer 2 Report

Thank you for the authors and the current manuscript has been greatly improved. My questions have been addressed.